# Antibody Responsiveness to Influenza: What Drives It?

**DOI:** 10.3390/v13071400

**Published:** 2021-07-19

**Authors:** Xia Lin, Fangmei Lin, Tingting Liang, Mariette F. Ducatez, Mark Zanin, Sook-San Wong

**Affiliations:** 1State Key Laboratory of Respiratory Diseases, Guangzhou Medical University, 195 Dongfengxi Rd, Guangzhou 510182, China; linxia618@163.com (X.L.); linfangmei@126.com (F.L.); 18671172019@163.com (T.L.); mark.zanin@gird.cn (M.Z.); 2IHAP, UMR1225, Université de Toulouse, INRAe, ENVT, 31076 Toulouse, France; mariette.ducatez@envt.fr; 3School of Public Health, The University of Hong Kong, Hong Kong, China

**Keywords:** influenza, antibody responses, seroconversion, immunity, vaccine

## Abstract

The induction of a specific antibody response has long been accepted as a serological hallmark of recent infection or antigen exposure. Much of our understanding of the influenza antibody response has been derived from studying antibodies that target the hemagglutinin (HA) protein. However, growing evidence points to limitations associated with this approach. In this review, we aim to highlight the issue of antibody non-responsiveness after influenza virus infection and vaccination. We will then provide an overview of the major factors known to influence antibody responsiveness to influenza after infection and vaccination. We discuss the biological factors such as age, sex, influence of prior immunity, genetics, and some chronic infections that may affect the induction of influenza antibody responses. We also discuss the technical factors, such as assay choices, strain variations, and viral properties that may influence the sensitivity of the assays used to measure influenza antibodies. Understanding these factors will hopefully provide a more comprehensive picture of what influenza immunogenicity and protection means, which will be important in our effort to improve influenza vaccines.

## 1. Introduction

### 1.1. Burden of Disease

Influenza viruses cause acute respiratory infections with significant mortality and morbidity in humans [1]. Annual seasonal influenza epidemics can result in three to five million hospitalizations and 290,000 to 650,000 respiratory disease-related deaths worldwide, with younger children and the elderly disproportionately affected [2]. An estimated 20% of children are infected annually by influenza [3], while 90% of influenza-associated deaths are due to underlying respiratory and circulatory complications in those aged 65 years or older [4]. The burden of disease is greater during pandemics involving novel subtypes for which population immunity is lacking. There have been at least four influenza pandemics in the last 100 years, the most severe of which was the 1918 pandemic that was estimated to have killed at least 50 million people [5]. Currently, two subtypes of influenza A viruses (IAVs), H1N1 and H3N2, and two lineages of influenza B viruses (IBVs), B/Yamagata and B/Victoria, are responsible for the seasonal influenza epidemics, with one or two strains dominating in a particular season or geographical location.

### 1.2. Antibody Responses to Influenza Viral Proteins

#### 1.2.1. Hemagglutinin (HA) and Neuraminidase (NA)

Hemagglutinin (HA) and neuraminidase (NA) are the two main antigenic proteins on the surface of IAVs and IBVs [6]. HA mediates binding to sialic acids on the host cells and facilitates fusion between the viral envelope and the host cell membrane [7]. A functional HA is a trimeric integral membrane protein, whose ectodomain consists of the globular head and the stem domain. The globular head contains the receptor binding sites and most of the human antibody responses are directed towards epitopes located here. Antibodies that target the globular head are immunodominant, affinity-matured, bind with high-specificity, and are generally neutralizing as they interfere with the binding of HA and sialic acids [8]. Due to this, the head is under the greatest immune pressure, resulting in continuous antigenic drift [9]. Immunity mediated by the HA globular head-antibodies following natural infection may even provide lifelong protective immunity against antigenically-related viruses that have not undergone antigenic drift. Antibodies targeting the globular head are also the main type of antibody detected by the classical hemagglutination-inhibition (HAI) assay [8]. Although HAI-assay does not strictly measure virus neutralization activity, HAI-antibody titers are generally well-correlated with virus neutralization titers and are considered to be an established correlate of protection in healthy adults [10].

In contrast to the relatively variable HA globular head, the stem region is more conserved amongst the different subtypes. Therefore, antibodies that target the HA-stem form the majority of the broadly neutralizing influenza antibodies (bnAbs) identified so far. Whilst they are produced at very low titers after infection compared to head-antibodies [11], they have been associated with protection in human challenges [12] and animal studies [13].

NA is a sialidase expressed as a homotetrameric glycoprotein spike on the viral surface. NA is divided into four domains: the cytoplasmic tail, transmembrane domain, stem domain, and the head which contains the enzymatic active site [14]. NA cleaves the sialic acid residues on the cell surface to promote the release of the newly formed virus particles as well as their motion through the respiratory mucus [15]. Antibodies targeting NA are previously thought to be unable to inhibit virus entry, and just prevent viral egress [11], but a recent study showed otherwise [16]. The contribution of NA-specific antibodies in reducing disease severity was recognized nearly fifty years ago during the 1968 A(H3N2) pandemic [17]. Epidemiological studies since then have shown that independent of HA, NA antibodies are also a predictor of immunity to influenza virus infections as they can limit or even prevent disease [18,19,20,21,22]. There is also evidence that NA vaccination prevents virus transmission in the guinea pig model [23]. Notably, the antigenic evolution of NA is slower than HA, thus antibodies targeting NA can potentially provide longer-term immunity compared to HA-antibodies [24]. In addition, NA-antibodies usually show broad cross-reactivity, and monoclonal antibodies that cross-react with NAs across influenza subtypes and even types have recently been identified [9,25]. Although currently licensed inactivated influenza vaccines generally contain sufficient residual amounts of NA to induce NA-antibodies [26], this response has not been systematically studied as the NA-content and tetrameric NA activity in vaccines are not standardized [27].

#### 1.2.2. Matrix 2 (M2) and Nucleoprotein (NP)

Another protein on the virus surface is the Matrix 2 (M2) protein, which is present at lower molar amounts compared to HA and NA but is expressed in large quantities on the surface of infected cells [28]. M2 is a tetrameric transmembrane protein that functions as a proton-channel and, whilst it is highly conserved across influenza subtypes, it is poorly immunogenic due to its low abundance and inaccessibility to antibodies [29,30]. However, its conservation makes it a promising target and, as such, the M2-ectodomain (M2e), has been developed as a subunit vaccine candidate [31]. In mice, M2e antibodies were shown to be protective, although the mechanism of protection appeared to be Fc-mediated [32,33]. Another conserved viral protein with reported protective function is the Nucleoprotein (NP), which is located within the viral envelope. Although NP has an important role in CD8^+^ T-cell mediated protection, antibodies targeting NP have been reported to provide some heterosubtypic protection in mice studies through non-neutralizing immune mechanisms [34]. The remaining viral proteins are internal and elicit relatively fewer antibodies with unknown significance for protection.

## 2. Immune Mechanisms of Protection

### 2.1. Virus Neutralization

The classical and most established correlate of protection against influenza virus infection is the presence of HAI-antibodies. Because HAI-antibodies recognize proteins on the virus surface and are generally well correlated with neutralization activity, its mechanism of action is postulated to be via direct binding and clearance of virus particles in the infected hosts. Based on the study conducted by Dobson et al., an HAI titer ≥ 1:40 was associated with a 50% reduction in infection risks in a healthy adult population [35]. This standard, considered to be the “seroprotective” threshold, has been adopted by both the U.S. Food and Drug Administration and the European Medicines Agency’s Committee on Human Use of Medicines for Influenza Vaccine Licensure. However, this seroprotective titer may not apply to young children or the elderly, since their immunological status likely differs compared to healthy adults [36,37].

Aside from the HA globular head, antibodies targeting other viral epitopes can also inhibit influenza virus infections in vitro and show protection in animal models. As described above, broadly-neutralizing influenza antibodies typically target conserved HA and NA epitopes, although may not be elicited in high titers after infection [38]. Dugan et al., recently showed that monoclonal antibodies targeting NA but not NP had neutralization activity and were protective in virus-challenged mice. However, their neutralization and protective potency were reduced compared to HA-globular head or HA-stem antibodies [16].

### 2.2. Fc-Mediated Mechanisms

The application of novel immunological approaches has identified a role for influenza antibodies in mechanisms of immune protection, other than virus neutralization. Non-neutralizing influenza antibodies can also mediate clearance of infected cells through Fc-dependent mechanisms namely, antibody-dependent cellular toxicity (ADCC), antibody-dependent cellular phagocytosis (ADCP), and complement-dependent cellular toxicity (CDCC) [39,40]. ADCC is primarily mediated through human FcγRIIIa found on natural killer (NK) cells, monocytes, and macrophages that release cytolytic enzymes when they recognize an IgG-bound immune complex on the surface of infected cells [41,42]. Aside from homologous strain reactivity, cross-reactive influenza ADCC antibodies can be detected in the sera of healthy individuals, individuals previously infected with the 2009 A(H1N1) pandemic strains, or those vaccinated with seasonal inactivated influenza vaccines [43,44]. Vaccination with 2014–15 seasonal influenza vaccine can induce HA-specific non-neutralizing antibodies, with strong ADCC activity even against antigenically drifted A(H3N2) viruses [44]. Seasonal vaccination of the elderly can also induce antibodies with strong ADCC activity, including against H5 and H7 subtypes [45]. Antibodies with ADCC activity that recognize internal viral proteins such as NP, M, and M2 have also been detected after influenza infection and vaccination [8], some of which have been shown to be protective in mouse studies [46].

ADCP is the antibody-mediated opsonization of infected cells expressing viral antigens on their surface by effector cells such as macrophages and neutrophils [47,48]. Antibodies with ADCP activity detected in post-infection human and macaque sera have been associated with reduction of infectivity in vitro and can be cross-reactive [49].

In CDCC, the complement proteins, which are soluble or membrane-bound factors that circulate in the blood and tissues, contribute to protection via direct clearance of pathogens or enhancement of adaptive immune responses. Influenza viruses can be neutralized via all three complement pathways, classical, alternative, or the mannose-binding lectin (MBL) pathway [50,51]. Although non-antigen-specific IgM (natural antibodies) can induce CDCC-mediated neutralization of influenza viruses [51], maximum benefits appeared to be elicited through antigen-specific antibody-mediated activation of both classical and alternative pathways [52]. As in ADCC, non-neutralizing ADPC and CDCC antibodies typically target non-HA1 epitopes and conserved proteins such as M2 [53,54,55,56]. These antibodies can mediate viral clearance either exclusively or through multiple Fc-mediated mechanisms [54,57].

In summary, non-neutralizing antibodies can be detected in the population, seemingly accumulate with age, and may contribute to protection against newly emerging influenza viruses, although the degree of in vivo protection in humans has not yet been formally established.

### 2.3. T-Cell Mediated Immunity

Cell-mediated immunity has also been shown to be a correlate of protection against influenza, particularly in reducing disease severity. T-cell mediated immunity mainly targets the more conserved internal viral proteins, such as NP, matrix protein 1 (M1), or the polymerase proteins (PA, PB1, PB2) [8], and as such, can potentially provide broader protection against different strains and subtypes. Both preexisting CD8^+^ and CD4^+^ T cells have been associated with reduced symptoms and viral shedding after infection in humans [58,59] and in a non-human primate challenge study [60].

At present, HAI-antibodies are best associated with protection from influenza virus infection and disease. However, as shown from human and animal studies [12,21], other non-HAI responses may also represent significant immune correlates of protection. These potential correlates are understudied, primarily due to the lack of standardized assays.

## 3. Signatures of Robust Antibody Response

### 3.1. Genetic Correlates of Antibody Response

Systems biology studies profiling human vaccine-induced transcriptomics responses have identified genes that predicted the immune response of a vaccine against the yellow fever virus [61,62]. Since then, transcriptomic profiling of the immune response to influenza virus infection and vaccination has contributed to our understanding of factors influencing robust antibody response [63]. Most of these studies were conducted using peripheral blood, which is easier to sample.

The strongest predictor of robust antibody response after influenza vaccination is the upregulation of the interferon (IFN) signaling genes (ISG), regardless of age or vaccines [64,65,66,67,68]. Strong induction of IFN gene expression positively correlated with the magnitude of the vaccine-induced antibody response to both live-attenuated influenza vaccine (LAIV) and trivalent inactivated influenza vaccine (TIV). Although a more rapid IFN-response in the peripheral blood was reported for TIV, it is unclear how much of this was influenced the route of vaccine administration [67]. Furthermore, the addition of squalene oil-in-water adjuvants to TIV also showed the same correlation between IFN-signature and the magnitude of HAI-titers in children and adults [68].

In addition to IFN-gene expression, genes that regulate B-cell proliferation have also been associated with a robust antibody response [69]. Among the differentially expressed genes induced after influenza vaccination, most are highly expressed in antibody-secreting cells [65] and dendritic cells [65,68]. Other markers of vaccine response included activation of genes associated with apoptosis function [70], membrane trafficking, and antigen processing [71].

Gene expression analysis in post-infection peripheral blood in a cohort of IAV-infected participants within 48 h of illness onset identified a total of 229 genes that correlated with development of HAI-titers. Similar to post-vaccination responses, most of the positively correlated genes were immune-related and associated with B-cell proliferation, while most of the negatively correlated genes were involved in programmed cell death pathways [72], likely due to the activation and proliferation of immune cells.

### 3.2. Cellular Correlates of Antibody Response

The production of high-affinity, durable antibody and B-cell memory responses, which require the initiation of a germinal center (GC) response in secondary lymphoid organs, is a multi-step process involving multiple innate and adaptive immune cells and cytokine signals. On a cellular level, studies have shown that early proliferation of plasmablasts or antibody-secreting cells (ASC) are a marker of subsequent antibody rise after vaccination [73]. Recent studies have also identified the circulating counterpart of the CD4^+^ T-follicular helper cells (cTfh) as a marker of antibody responsiveness after influenza virus vaccination [74]. Identifying early cellular correlates of antibody production after infection is more challenging due to the difficulties in obtaining early blood samples, although we recently identified actively proliferating CD4^+^ T-cells as a cellular correlate of subsequent seroconversion [75]. Activated cTfh-1 has also been found to be a predictor of robust antibody response in SARS-CoV-2 infections [76,77].

In summary, host signatures associated with robust antibody response are largely conserved at a global level in the peripheral blood between influenza vaccination and infection in that they stimulate a strong IFN-driven pro-inflammatory response, initiate antigen presentation, and activation of CD4^+^ T-cells and B-cells early in the course of exposure. A summary of representative studies and their key findings are shown in Table 1.

## 4. Antibody Non-Responsiveness after Vaccination

Although there are many licensed formulations for seasonal influenza vaccines, the two most established are the Inactivated Influenza Vaccine (IIV) and LAIV, which contain either three (trivalent, TIV) or four (quadrivalent, QIV) strains and are updated annually for the Northern or Southern Hemisphere based on the strains most likely to predominate in the upcoming influenza season. TIV and QIV contain two subtypes of IAV, A(H1N1) and A(H3N2), TIV contains one IBV, which was predicted to be the dominant circulating lineage for that year, while QIV will contain IBVs from both lineages (Yamagata and Victoria). LAIV is based on an “attenuated”, cold-adapted virus strain expressing the HA and NA of the recommended vaccine strains. LAIV replicates optimally at 25 °C, and not at temperatures higher than 35 °C, which is the temperature of the respiratory tract. This limited replication capacity of LAIV can stimulate both local humoral and cell-mediated immunity, unlike IIVs which do not induce a robust cell-mediated response. The immunogenicity of a vaccine is typically based on seroconversion rates or the development of seroprotective titers (HAI titer ≥ 1:40) [83].

Influenza vaccine immunogenicity is well-studied and many reviews and meta-analyses are available that summarize these findings. Not surprisingly, immunogenicity is dependent on age, disease, immune status, and pre-existing influenza immunity, which will be discussed in further detail below.

On the vaccine side, vaccine manufacturing processes, batch-to-batch variation, and vaccine virus strains can impact immunogenicity [84]. For example, a review of the seroconversion rates elicited by vaccines against A/Viet Nam/1203/2004 (H5N1) in vaccine trials showed that vaccines with equivalent amounts of HA but from different manufacturers elicited seroconversion rates ranging from 0% to 43% [85]. In addition, vaccines against avian influenza strains are notoriously poorly immunogenic in humans [85], requiring higher doses or adjuvants to induce acceptable seroconversion rates [86,87]. Amongst the different avian influenza vaccines that proceeded to Phase II trials, those against A(H7N9) viruses performed the best [88,89], although not to the level seen with seasonal influenza vaccines. This is not due to the inherently poor immunogenicity of avian strains, as A(H7N9) and TIV induced comparable titers in naïve ferrets, suggesting that insufficient immune priming might be a factor for the poor responses seen in humans. We and others have reported that physical composition and protein integrity may play a role in the antibody response against a vaccine [85,90].

Another significant example of the impact of influenza strain on vaccine immunogenicity is the poor performance of the A(H1N1) component of LAIV, which was the 2009 A(H1N1) pandemic virus [91]. Between 2010 to 2016, the US CDC vaccine efficacy (VE) studies reported poor VE against A(H1N1) provided by LAIV. Since its introduction into LAIV, the 2009 A(H1N1) pandemic component did not appear to replicate nor induce any HAI-antibodies, unlike the A(H3N2) or IBV components of the vaccine [92]. Many potential factors have been ascribed to this observed decreased effectiveness, including the poor replicative fitness and thermal stability of the early vaccine strain. A change to a strain with higher replicative ability based on in vitro assays for the 2017/2018 influenza season resulted in improved immunogenicity, shedding, and VE in the UK where it was recommended for use [93,94].

## 5. Antibody Non-Responsiveness after Infection

Identifying the correlates of a robust antibody response is easier within the context of vaccination due to a temporally defined pre and post-antigen exposure that facilitates sample and data collection. In the case of influenza, seroconversion would typically be based on HAI-antibody titer increases, which was the cornerstone of influenza diagnosis before the advent of molecular techniques. However, the use of both PCR-diagnosis and serology in large seroepidemiological and human challenge studies has indicated that influenza virus infections do not always result in seroconversions (Table 2).

Early LAIV trials and recent human challenge studies have been useful in delineating the association between pre-existing immunity and infection doses with subsequent antibody responses. In general, a larger infection dose is associated with a higher seroconversion rate and higher antibody titers [59,95,96], although this relationship is not linear past a specific threshold dose. Memoli et al. showed that an infectious dose greater than 10^6^ tissue-culture infectious dose-50 (TCID50) of the H1N1pdm09 strain induced seroconversion in 85% of volunteers, while lower doses induced seroconversion in only 20% of volunteers. Notably, a similar study using an A(H3N2) virus found seroconversion in only 29% (10/35) of inoculated volunteers. It is also worth noting that the cohort had high serum N2-antibody titers, which could have been a contributing factor in the less efficient challenge compared to the H1N1pdm09 challenge study [97]. Studies using LAIV, in which shedding of vaccine virus is a measure of successful infection, have shown that preexisting immunity, including mucosal antibodies, can prevent viral shedding. Although non-shedding does not preclude induction of systemic HAI-antibodies, positive shedders tended to have a larger fold increase in the subsequent antibody response [98,99], which was in line with animal studies [100].

How does this translate to the community setting? Prior to the emergence of coronavirus disease 2019 (COVID-19), relatively few community surveillance studies that combine serologic with a molecular diagnosis were conducted. A seroepidemiologic study conducted in Singapore during the 2009 A(H1N1) pandemic reported that 80% of laboratory-confirmed individuals eventually seroconverted [101], while we observed only 32% of seroconversion in a New Zealand cohort during an A(H3N2) and IBV-dominated season [75]. In the latter study, seroconversion was more frequently observed in hospitalized patients, consistent with data from human challenge and COVID-19 studies [102] that suggest that severity of infection may correlate with the induction of antibody responses [95]. A recent multicenter study on the burden of respiratory infections in infants showed that 23% of influenza-positive infants did not seroconvert to the tested strains by HAI or microneutralization assays [103].

Overall, the correlation between the initial infection dose, disease severity, and antibody development is a challenge to address in any naturally-acquired infection studies due to difficulties in determining the time of infection. Although the time and infection doses are predetermined in LAIV and human challenge studies, for ethical considerations, the viruses used are usually attenuated, or inoculated at doses that do not induce significant symptoms. As such, the relationship between the initial infection dose, symptom severity, and the subsequent downstream antibody response in humans is not well-established and can only be inferred from animal studies.

## 6. Biological Factors That Influence Antibody Non-Responsiveness

### 6.1. Age: Immunosenescence, Frailty

In children, poor vaccine responses have been attributed to a lack of priming, hence two doses have been recommended for children younger than 12 years of age in many countries [106]. However, factors underlying poor vaccine responses in the elderly, considered to be 60 years and above, are more complicated. A quantitative review of 31 studies using IIV found lower seroprotection and seroconversion rates in the elderly compared to adults, although, notably, this was more evident for A(H3N2) compared to A(H1N1) and IBV, with heterogeneity observed across the different studies [107]. However, a number of studies also reported no differences [108] or better responses compared to adults [109].

Post-vaccination antibody response failures in the elderly have been classified either as failure to mount antibody response within four weeks of vaccination (primary failure) or inability to sustain a post-vaccination antibody response (secondary failure) [109]. In these cases, the decline in vaccine-induced immune response in the elderly has been in part, attributed to the age-associated decline in innate and adaptive immune response, or immunosenescence [110,111]. Specifically, the expression of co-stimulatory molecules CD80 and CD86 on monocytes [112], and their downstream receptors CD28 on T-cells have been implicated in the impaired influenza vaccine responses in the elderly. Intrinsic defects in the B-cell compartment, including the impaired activity of the activation-induced cytidine deaminase (AID) that is required for class-switching memory B cells, decreased diversity and plasticity of the B-cell receptor repertoires, and shorter B cell telomere length may also contribute to the reduced immune response in the elderly [113,114,115,116,117]. Moreover, older adults develop fewer de novo immunoglobulin gene somatic mutations, which suggests a limited capacity to respond to novel antigens [118].

In addition to immunosenescence, frailty, a geriatric syndrome which is characterized by increased vulnerability to adverse health outcomes and multi-system dysregulation, has also been proposed to affect vaccine-induced antibody response in the elderly [119]. Some studies have demonstrated that vaccine-induced antibody response and vaccine effectiveness decreased as frailty increased [120,121], others reported no differences [122,123,124], while Loeb et al., using an approach that treated frailty as a continuous variable, found that antibody responses were positively associated with frailty after high-dose vaccination [125]. These inconsistent results may highlight the difficulty in classifying frailty, particularly at an immunological level.

### 6.2. Prior Immunity: Repeat Vaccination, Immune Priming, and Imprinting

It is now becoming clear that pre-existing influenza immunity can influence the antibody responses after influenza vaccination and infection in different ways. One consistent observation across vaccine trials is that while higher preexisting HAI-titers to the vaccine strain correlated with greater odds of achieving seroprotective titer, it is inversely correlated to seroconversions [3].

The constant strain updates to influenza vaccine composition and the recommendation for annual vaccination represents a particularly unique challenge for influenza. After repeated influenza vaccination, the serological response to recent vaccination may decrease. Repeat-influenza vaccination can result in higher baseline titers and lower magnitude fold changes, particularly when the vaccine strains were unchanged [126,127]. Importantly, reduced immunogenicity after repeated vaccination did not appear to be associated with reduced protection [128], likely because seroprotective titers were already present or protection was mediated by non-HAI-antibodies.

As early as 1960, Thomas Francis Jr and his colleagues observed that the antibody response to influenza virus strains in childhood predominated during subsequent infections by antigenically related strains, a phenomenon he called original antigenic sin (OAS) [129]. Since then, studies have shown that the highest magnitude antibody responses are elicited to strains circulating within the first 10 years of life, and antibodies are elicited in a hierarchical manner to subsequently circulating strains, or “antigenic seniority”. This suggests that early-life exposure will determine the pool of memory B-cells available to be recalled during subsequent exposures. Indeed, a recent study demonstrated that 60% of monoclonal antibodies elicited after influenza infection displayed equal or stronger affinity to childhood strains, indicating a strong bias of recall memory response from childhood exposures [16]. The effects of OAS can be reflected in the HAI and NA inhibiting (NAI)-antibodies as well as the HA-IgG antibodies responses [130].

How do pre-existing influenza antibodies influence the subsequent profile of antibody response? A recent informative study by Dugan et al. characterized the monoclonal antibodies derived from plasmablasts induced after infection and vaccination and found that only an average of 29% of antibodies elicited after an infection has in vitro virus-neutralizing activity, compared to 80% of antibodies that were elicited after vaccination. The non-neutralizing antibodies induced after infection target more conserved epitopes such as HA-stem, NA, and NP, and other as yet unidentified epitopes. Interestingly, they noted differences in the H1N1 and H3N2 responses; H3N2 were more likely to elicit cross-reactive antibody responses compared to H1N1, which they attributed to the more rapid evolution of H3N2 viruses [16]. Andrews et al. showed that individuals with low preexisting serum HAI-titers generated more stem reactive plasmablasts after vaccination, while those with HAI titers >40 were associated with the generation of more HA-head targeting plasmablasts [38]. Incidentally, antibodies targeting the stem domains are most enriched during infection or vaccination with antigenically-shifted strains or strains with divergent HA-globular heads. Collectively, these studies demonstrate that each new exposure demonstrates a bias in recalling pre-existing memory B-cells, which with repeated exposures, may be enriched against conserved epitopes that have less potent neutralizing activity. Incidentally, however, pre-existing antibodies can also bind to such epitopes (epitope-masking), reducing antigen availability for subsequent antigen presentation [131]. It is important to note, however, that OAS has not been shown to impede the development of de novo antibodies, as greater increases in antibody titers are typically detected against the immunizing antigens [132,133].

The number of exposures and age are therefore important determinants on the influenza antibody landscape, used here to describe the totality of influenza antibody response. The elderly (≥65 years old) will have the most influenza exposures, and correspondingly, have been shown to possess the highest baseline levels of HA-stem and NA-antibodies compared to other age groups [134,135]. Indeed, using an influenza virus protein microarray, Meade et al. found that children under 6 years of age had a narrow lgG and lgA antibody response while adults showed a broad recall response [136]. Age can also influence the antibody dynamics to HA and NA in a strain-specific manner; adults are more likely to show either a HA- or NA-dominant response compared to children after influenza A, but not influenza B virus infections [137]. Since non-HAI antibodies have been identified as additional correlates of protection, it is important to therefore understand their recall dynamics. Efforts are currently underway to incorporate these assays as standard endpoints in influenza studies [138,139].

### 6.3. Sex-Based Differences

Although sex and gender can both influence influenza disease pathogenesis and the immune response, here we discuss only the role of sex as a biological construct and its effect on the influenza antibody responses. Epidemiologic data revealed females and males had different morbidity and mortality against seasonal, avian, and pandemic influenza in an age-dependent manner [140,141], suggesting a potential sex-based bias in the host immune response. Sex-based differences in antibody response have been better described in post-influenza vaccination antibody responses. Several studies found that females across different age groups developed stronger antibody responses following seasonal influenza vaccination compared to males [140,142,143,144]. Vaccine effectiveness was also greater in females than males, especially against influenza A(H3N2) and IBV (Victoria) strains [145]. Higher concentrations of inflammatory cytokines, such as Granulocyte macrophage-colony stimulating factor (GM-CSF), Interleukin (IL)-5, and IL-6 were detected after influenza vaccination in females compared to males [146,147]. Transcriptomics studies of gene expression after vaccination revealed greater expression of genes related to the immune response in females within 24 h, although it was not indicated whether this cohort demonstrated different post-vaccination antibody responses [148]. Older females had significant differences in their NK, T- and B-cells gene expression in which the latter correlated with the development of higher memory B-cells responses as measured by ELISPOT assay, compared to males [149].

The underlying sex-based differences in antibody response have been attributed primarily to the influence of sex hormones and genetic factors. For example, immune-regulatory genes such as *IL13rα2* and *Tlr7* are encoded on the sex-chromosomes [150,151], and high testosterone levels have been reported to suppress vaccination responses in males [146], although this has also been disputed [152]. A positive correlation between concentrations of estradiol and the antibody response has been observed in females [146,147]. The influence of sex hormones during the different life stages could also explain age-specific immune responses [147,153].

Small animal models have been used to recapitulate these sex-based differences. Female mice show greater CD4^+^ T and B-cell proliferation and developed more robust neutralizing antibody and total IgG responses compared to male mice following influenza vaccination [154,155,156,157,158,159]. Estradiol has also been reported to restore antibody responses in a postmenopausal mouse model immunized with a A(H1N1) vaccine, suggesting an underlying role for sex steroids in the sex-base differences observed in antibody responses [160]. An exception is that adult human males (18 to 64 years old) showed a greater proportion of high-avidity antibodies after monovalent 2009 A(H1N1) pandemic vaccination compared to age-matched females [160], although this observation has not been replicated in mice [156,157,158].

There is little evidence on differences in the sex-based antibody response after influenza virus infection in humans although, female mice generated more neutralizing antibodies and total anti-influenza antibodies than male mice after infection with A(H1N1) or A(H3N2) [161]. Moreover, female mice developed greater cross-protection against heterosubtypic influenza virus compared to males [161]. Incidentally, female mice are reported to be susceptible to more severe infections compared to male mice [162,163].

The evidence remains controversial on sex-based differences in the immune response to influenza in humans. Some studies have found differences at cellular but not soluble antibody levels; that is, sex-based differences in plasmablasts and B-cells responses but not in HAI-titers, suggesting potential sex-based differences in the proliferation and secretory response of the humoral immune response [149,164]. Whilst some epidemiological and animal studies have provided convincing evidence, the underlying mechanisms and relative contribution of sex within the context of other factors to antibody responses need to be resolved.

### 6.4. Genetics

While host-genetic factors have been associated with influenza pathogenesis [165], less is known about its direct impact on antibody responses. A recent study found that genetic polymorphism in Interferon-induced transmembrane protein 3 (IFITM3), were associated with the magnitude of the antibody response after seasonal influenza vaccination [166]. Individuals with *IFITM3* rs12252-C/C genotype had lower post-vaccination seroconversion rates compared with C/T and T/T donors. Poorer antibody responses were also detected in the *Ifitm3*−/− mouse model compared to wild-type. *IFITM3* rs12252-C/C were also reported to be associated with severe influenza, presumably due to a truncated IFITM3 protein with reduced antiviral potency [167,168], although others found no such association [169,170]. However, the *IFITM3* rs12252 allele is only one of several elements in the interferon signaling pathway that has been associated with influenza clinical severity [171,172]. Given that the human interferon-system is such a critical modulator of the immune responses and is the common element that was associated with robust antibody responses after infection or vaccination, it would be important to explore further the contribution of these genetic polymorphisms to antibody responses.

### 6.5. Chronic Infections

#### 6.5.1. Cytomegalovirus (CMV)

Cytomegalovirus (CMV) is a β-herpesvirus that infects between 40–100% of the adult population worldwide, establishing lifelong latency. Seropositivity within the population increases with age, with most studies reporting at least 60% seropositivity in those aged 50 years and above [173].

CMV is postulated to modulate host immunity via two mechanisms; a direct effect of the viral proteins, and, inducing a state of chronic inflammation during reactivation. Aside from inducing the expression of human IL-10 during infection, the CMV genome encodes cmvIL10, a human IL-10 homolog whose expression may differ during acute or latent infections. Like the human IL-10, cmvIL10 has a broad range of immunomodulatory functions, particularly on myeloid cells, which also serve as a latent reservoir for the virus. It has been reported to inhibit the maturation and function of DCs and down-regulate the expression of MHC Class II on myeloid cells, all of which are detrimental to antigen presentation [174,175]. Reactivation of CMV can lead to a state of chronic inflammation, postulated to lead to “exhaustion” of various immune cell populations [176]. CMV infection in humans is also linked to increased concentrations of immune-modulatory cytokines in the blood, such as TNFα, IL10, and IL-6 [177,178,179].

CMV serostatus also has implications for B and T-cell populations and their responses to vaccination. A negative association has been found between CMV-seropositivity and predictive biomarkers of optimal vaccine responses in B-cells, namely switched memory B cells and activation-induced cytidine deaminase (AID) [180,181,182,183]. Further, CMV-seropositivity has a large effect on CD4^+^ and CD8^+^ T-cell subsets at all ages in healthy individuals [184]. CMV infection primarily results in the accumulation of late-differentiated memory T-cells, both in the CD4^+^ and CD8^+^ T-cell lineage [185,186,187], which have limited capacity to respond to novel antigens. The late-stage differentiated memory T-cells that accumulate in the elderly, especially the CD8^+^ T-cells, are frequently CMV-specific. Their eventual loss may be associated with incipient mortality [188]. A similar phenomenon has also been observed in CD4^+^ T-cells, where CMV-associated accumulations of late-stage memory cells and reduced naive cells have also been reported [189]. Many studies have shown that these factors are associated with a poor immune response to influenza [190,191,192,193].

Some studies have shown that CMV-seropositivity negatively correlates with post-influenza vaccination antibody responses [180,194,195,196], while others found no correlation [195,197,198]. One study described an age-specific effect, whereby CMV infection enhanced post-influenza vaccination responses in the young but not the old [199]. Latent CMV can be reactivated during conditions of immunosuppression or inflammatory stress due to infections or critical illness [200]. Reactivation of CMV and Herpes simplex virus type-1 (HSV-1) have been reported in critically ill A(H7N9) patients [201,202]. Given its importance in modulating the immune response, how common herpesvirus reactivations are and the consequences on host immunity have not been directly examined in the context of influenza virus infections, particularly in immunocompetent hosts. The lack of consensus on the significance of CMV to influenza vaccination and infection is attributed in part to the difficulty in detecting active CMV infections, and the functional consequence of being CMV-seropositive [203]. More research is needed to clarify the relationship between CMV and influenza virus infection and immunity.

#### 6.5.2. Human Immunodeficiency Virus (HIV)

As with CMV, there are conflicting accounts on the relationship between human immunodeficiency virus (HIV) infection and influenza immunity. Antibody responses to influenza are observed to be relatively poor in HIV-positive patients [204,205,206,207,208,209], likely due to the presence of low numbers of CD4^+^ T-cells and viremia [207,210,211]. Overall, there is limited data on the clinical effects of influenza vaccination in the context of HIV, however, it is important to address this as HIV patients are a high-risk group for influenza.

## 7. Technical Factors: Choice of Assays, Antigens, and Samples

Whilst biological factors can cause poor antibody responses after influenza virus infection and vaccination, virological or technical factors can lead to reduced assay sensitivity or false-negative or false-positive results. Due to the importance of HAI-antibodies and the relative ease of use, HAI-assays have been the serological workhorse in the field. However, HAI-assays have technical limitations. For example, the utility and sensitivity of this assay depend on the agglutination of red-blood cells (RBCs) by the virus. Adaptive HA mutations can alter sialic acid binding specificity and thus binding to RBCs, impacting the results of HAI assays. This can be overcome by using RBCs from other species than chicken, which are most commonly used, owing to species differences in the expression of sialic acids on RBCs [212]. For example, horse RBCs can increase the detection sensitivity of antibodies against avian influenza viruses [213,214], while guinea pig and human O-type RBCs are recommended for seasonal influenza viruses [215]. Incidentally, most recently isolated A(H3N2) influenza viruses have HA mutations abrogating RBC binding, necessitating the adoption of alternative assays such as microneutralization or foci-reduction neutralization assays that do not rely on RBCs [216]. Although not as commonly used, single-radial hemolysis (SRH) is are also considered a gold standard in influenza immunogenicity evaluations by some regulatory bodies. Because it relies on complements to cause hemolysis, SRH can be used to detect complement activating influenza antibodies. These serological assays, along with ELISAs are capable of detecting influenza antibodies with different functional properties and generally show good correlations to one another in a robust antibody response [217,218].

The method of antigen preparation can also influence serological observations. For example, using ether-treated IBVs as antigens in the HAI assay resulted in higher detection sensitivity, albeit at slightly lower specificity, compared to non-treated antigens [219]. Egg-grown influenza viruses used in the production of most influenza vaccines can also be antigenically distinct from cell-grown viruses, which are more similar to circulating strains [220] since egg-grown viruses can acquire egg-adaptive mutations that impact antigenicity [221,222].

It is becoming evident that the different classes of influenza antibodies beyond those detected by HAI-assays are functionally important and should also be considered as part of influenza antibody responses. Our study in New Zealand found that the inclusion of NAI-assay into the serodiagnosis platform increased the detection sensitivity of capturing influenza cases in the seroepidemiologic study [223]. The commonly used method to measure NAI antibodies is the enzyme-linked lectin assay (ELLA), which was originally developed by Lambre et al. [224]. It measures the degree of serum inhibition of NA-enzymatic activity using fetuin as a substrate. Compared with the traditional thiobarbituric acid (TBA) method described by Webster and Laver [225], this method offered the advantages of being scaleable, safe, and specific [226]. Another NAI assay relies on the enzymatic cleavage of smaller substrates; either the fluorescent 2′-(4-Methylumbelliferyl)-α-D-*N*-acetylneuraminic acid (MUNANA) assay [227] or the chemiluminescent NA-STAR assay [228]. Because of the small substrates, NAI-antibodies detected using the MUNANA or NA-STAR assays are thought to only bind near the enzyme’s active site.

It should also be mentioned that new assays are being developed to detect influenza antibodies. One such approach is the use of pseudoviruses bearing target influenza antigens. There are now many systems employed to produce these pseudoviruses [229]. The major advantages of using pseudoviruses are its safety, which is particularly relevant when expressing antigens from highly pathogenic avian influenza viruses, and its ability to express specific peptides, such as the HA-stem and M2. This enables the detection of antibodies targeting these epitopes that are normally inaccessible on wild-type viruses. The challenge remains in validating and standardizing these assays to current serological standards as they have shown variable performance in several comparative studies [218,230].

Therefore, the choice of an assay to measure these antibodies are not only qualitatively important, but can also reflect the dynamics that underlie the human immunological memory of influenza viruses. Finally, sampling technicalities could also impact the detection sensitivity for any antibody response. Appropriate time of sampling, sampling methods, and sample treatments could also affect the sensitivity in detecting seroconversion events [231].

Serum is considered the best specimen of choice for serologic assays. Compared to serum, plasma contained anticoagulants that can sometimes interfere with antibody-antigen interaction. HI titers were higher against influenza B in plasma, causing the overestimation and underestimation of the seropositive rates [232]. In general, however, there is a high degree correlation for HI and neutralization assays against influenza A between serum and plasma, which indicated that plasma can be used as an alternative specimen of choice for these assays, where convenient or necessary [232,233]. Appropriate sample treatment to remove non-specific inhibitors in human and animal sera is also crucial for accurate results. For HI assay, several treatment protocols are available although none of it worked universally across strains and species [234]. Treatment with receptor destroying enzyme (RDE) from *Vibrio cholerae* and heat-inactivation is presently recommended as part of the standard WHO HI protocol [215]. For an in-depth review of the serologic assays for influenza, we recommend reference [235].

## 8. Conclusions

Studying the immunogenicity of the influenza virus is complicated due to the large diversity of antigenic variants present in nature and our constant exposure to it, either through natural infection or vaccination. We have attempted to provide an up-to-date, although by no means comprehensive overview of factors that may influence influenza antibody responses and our ability to measure it (summarized in Figure 1). The biological factors described here may not be specific to influenza antibody responses and often do not occur independently. Some of the factors may even converge in a person’s lifetime, i.e., chronic infections, number of exposures, and advancing age. How such factors interact and importantly, their relative contributions to the immunological response to influenza during infection and vaccination should be considered. Furthermore, recent advances demonstrating the importance of looking beyond HA-based immunity will likely change serological standards in influenza studies. This will provide a more comprehensive picture of what influenza immunogenicity, antibody responses, and protection mean. Such knowledge is vital to the design of improved vaccines or targeted vaccination programs against influenza.

## Figures and Tables

**Figure 1 viruses-13-01400-f001:**
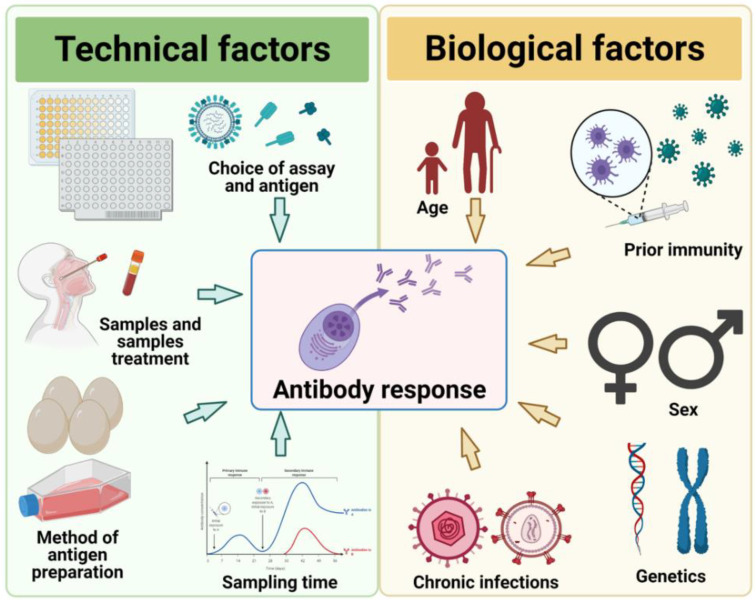
An overview of the biological and technical factors discussed in this review.

**Table 1 viruses-13-01400-t001:** Representative studies on the genetic and cellular correlates of robust antibody responses after influenza vaccination or infection.

		Cohort	Formulation/Strain	Season	Study Design	Findings	Reference
Genetic correlate	Vaccination	Children aged 12–35 months, *N* = 85	LAIV and TIV	2006–2007	Transcriptional profiling by microarray of whole blood RNA at Day 7 post-vaccination.	Type 1 interferon-stimulated gene (ISG) was upregulated, particularly robust in LAIV recipients although no correlation to antibody data was available.	[78]
		Children aged 6 months to 14 years, *N* = 37	LAIV and TIV	2010–2011	Blood samples were collected on day 0 before vaccination and on days 1, 7, and 30 after vaccination to measure gene expression profiles.	TIV had a more rapid IFN-response compared to LAIV. The overexpression of IFN genes in TIV and LAIV was correlated with H3N2 antibody titers.	[67]
		Children aged 14–24 months, *N* = 90	TIV and adjuvanted TIV	2012–2013	Transcriptional profiling by microarray of whole blood RNA at Day 1, 3, 7, and 28 post-vaccination.	TIV with adjuvant showed gene differences in IFN genes, dendritic, and monocyte cells, which was correlated with the antibody response.	[68]
		Adults, males aged 18–40 years, *N* = 119	TIV	2008-2009	Microarray analysis of peripheral blood samples before and on days 1, 3, and 14 post-vaccination.	Upregulation of IFN genes and antigen presentation pathways was associated with higher vaccine-induced antibody response.	[64]
		Adults aged 18–50 years, *N* = 67	LAIV and TIV	2008–2010	Microarray analyses of the gene expression profiles of PBMC at baseline, and days 3 and 7 post-vaccination.	Molecular signatures can be used to predict later antibody responses. Most of the genes induced by influenza vaccination are highly expressed in antibody-secreting cells.	[65]
		Adults aged 18–40 years, *N* = 119	TIV	2008–-2009	Global transcript abundance analysis of peripheral blood RNA specimens before and at days 1, 3, and 14 post-vaccination.	Membrane trafficking and antigen processing were associated with the immune response to the vaccine.	[71]
		Adults aged 18–45 years, *N* = 60	TIV	2012–2013	Transcriptomic analysis of blood samples at days 0, 1, and 21 post-vaccination.	Serum levels of CXCL10 were correlated with T cell and antibody responses after vaccination.	[79]
		Adults and elderly aged 20 to >89 years, *N* = 91	TIV	2008–2009	Whole-blood microarray analysis of gene expression at days 0 and 28 ± 7 post-vaccination	Genes involved in apoptosis were positively associated with vaccine-induced antibody response.	[70]
	Infection	Adults aged 18–49 years, *N* = 58	Influenza A and influenza B	2009–2011	Peripheral blood gene expression profiling at Day 0, 2, 4, 6, and 21 post-infection.	Influenza virus infection caused greater magnitude and longer duration of upregulation of interferon signaling pathway genes.	[72]
Cellular correlate	Vaccination	Adults aged ≥18 years, *N* = 44	MF59-adjuvanted H5N1	A/Vietnam/1194/2004 (H5N1), given as part of a trial with 3-study arms	PBMC was collected at baseline and 3 weeks post-vaccination.	Expansion of ICOS + IL-21 + CD4^+^ T cells was an early marker of antibody response.	[80]
		Adults, *N* = 49 and children, *N* = 20	TIV	2009–2012	PBMC was collected at baseline and day 7 post-vaccination	ICOS + CXCR3 + CXCR5 + CD4^+^ T cells correlated with antibody response induced by memory B cells.	[81]
		Adults aged 30–40 years, *N* = 28 and elderly aged ≥65 years, *N* = 35	TIV	2012–2013	PBMC was collected on days 0, 7, and 14 post-vaccination.	Circulating Tfh cells predicted antibody response in young but not elderly.	[82]
	Infection	Children and adults, aged 0–90 years, *N* = 16	Influenza A and influenza B	2013–2015 in New Zealand	PBMC collected immediately and at least 14 days after enrollment	CD4^+^ T-cells proliferation and greater inflammatory monocytes depletion was associated with HAI-seroconversion	[75]
		Children, aged 2 months to 34 years, *N* = 19	Influenza A and influenza B	2009–2013 in Memphis, TN	Nasal washes collected longitudinally upon enrollment	Inflammatory monocytes depletion was associated with subsequent production of nasal mucosal IgA and IgG	

**Table 2 viruses-13-01400-t002:** Seroconversion events reported in human influenza challenge studies, representative trials with live-attenuated influenza vaccines (LAIV), and seroepidemiologic studies.

Type of Study	Study	Study Design	Subtype	No. Infected	No. Seroconversion (%)	References
Human Challenge	Validation of the wild-type influenza A human challenge model H1N1pdMIST: An A(H1N1) pdm09 dose-finding investigational new drug study	10^3^ to 10^7^ TCID_50_ ^a^	A/California/04/2009 (H1N1)	46	29/46 (63%)	[95]
Human Challenge	A Dose-finding Study of a Wild-type Influenza A(H3N2) Virus in a Healthy Volunteer Human Challenge Model	10^4^ to 10^7^ TCID_50_	A/Bethesda/MM1/2011 (H3N2)	37	10/35 ^b^ (29%)	[97]
Human Challenge	Characterization of a wild-type influenza (A/H1N1) virus strain as an experimental challenge agent in humans	10^4^ to 10^6^ TCID_50_	A/California/04/2009 (H1N1)	29	14/29 (28%)	[96]
Human Challenge	Preexisting influenza-specific CD4^+^ T cells correlate with disease protection against influenza challenge in humans	10^3^ to 10^6^ TCID_50_	A/Wisconsin/67/2005 (H3N2, cell grown)	17	7/14 ^b^ (50%)	[59]
			A/Brisbane/59/2007 (H1N1, egg grown)	24	9/24 (38%)	
Human Challenge	Landscape of coordinated immune responses to H1N1 challenge in humans	3.5 × 10^6^ to 7 × 10^6^ TCID_50_	A/California/04/2009 (H1N1)	35	20/35 (57%)	[104]
LAIV trial	Evaluation of A/Alaska/6/77 (H3N2) cold-adapted recombinant viruses derived from A/Ann Arbor/6/60 cold-adapted donor virus in adult seronegative volunteers	1.5 × 10^4^ TCID_50_	A/Alaska/6/1977 (H3N2, wild-type)	8	7/8 (75%)	[105]
		3.2 × 10^7^ to 5 × 10^7^ TCID_50_	A/Alaska/6/1977 (H3N2, cold-adapted clones)			
LAIV trial	Dose-response of A/Alaska/6/77 (H3N2) cold-adapted reassortant vaccine virus in adult volunteers: role of local antibody in resistance to infection with vaccine virus.	1.5 × 10^4^ TCID_50_	A/Alaska/6/1977 (H3N2, wild-type)	8	7/8 (75%)	[98]
		3.2 × 10^7^	A/Alaska/6/1977 (H3N2, cold-adapted clone, CR29)	24	50%	
		3.2 × 10^6^		15	40%	
		3.2 × 10^5^		15	20%	
		3.2 × 10^4^		12	8%	
Seroepidemiologic	2009 influenza A(H1N1) seroconversion rates and risk factors among distinct adult cohorts in Singapore	Observational Cohort	A/California/04/2009 (H1N1)	56	45 (80%)	[101]
Seroepidemiologic	Activated CD4^+^ T-cells and CD14^++^ CD16^+^ monocytes correlate with antibody response following influenza virus infection in humans	Observational Cohort	Influenza A and B	66	21 (32%)	[75]
Seroepidemiologic	Underdetection of laboratory-confirmed influenza-associated hospital admissions among infants: a multicentre, prospective study	Observational Cohort	Influenza A and B	254	196 (77%)	[103]

^a^ TCID_50_: 50% tissue-culture infectious dose. ^b^ Participants were lost to follow-up.

## Data Availability

Not applicable.

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
