# Peer review of "Antibody Responsiveness to Influenza: What Drives It?"

_viruses, 2021, doi:10.3390/v13071400_

Round 1

Reviewer 1 Report

Xia Lin and colleagues present a review of the literature surrounding antibody responses generated to influenza A and B viruses of human importance.  This is a very well executed review, with extensive and useful review and citation of the relevant literature that would be useful to the readers.  Given the drop in detection of influenza viruses across the globe as a result of the SARS-CoV-2 pandemic, review of influenza literature is crucial to prevent complacency in research on this important virus.

I have listed minor edits as well as general comments below for the author’s consideration.

Line 49: “majority of antigenic sites” is too general, the authors should be more specific in stating that the majority of antibody responses in humans are directed to antigens in the globular head.

Line 53-54: “Immunity mediated by the HA globular head…” This statement is slightly misleading in the current wording.  Please be more specific such as “may even provide lifelong protective immunity against antigenically-related viruses that have not undergone antigenic drift.”

Line 59: The authors should mention HAI correlates of protection here, as despite its drawback the HAI assay is still gold standard, as referenced on line 106 [33]

Line 71 “not generally neutralizing”, this could be clarified by saying “Antibodies targeting NA are not generally able to inhibit virus entry, rather they have been shown to limit viral egress”. This clarifies that an antibody to NA is not neutralising, but inhibiting enzymatic activity which has an impact on a different part of the virus life cycle.

Line 82: The authors should include paper on tetrameric NA, therefore NA quality being crucial for immune responses. Eg. https://www.ncbi.nlm.nih.gov/pmc/articles/PMC5885027/

Line 101, “virus neutralization” as a subtitle is deceiving when talking about HAI research

Line 177 “immune correlates of robust antibody response” is not well used in the literature and could be misleading. This should possibly be changed to “factors influencing robust antibody response” or a similar type of statement excluding “immune correlates”

Line 183 to 185: The authors should consider mentioning the delivery route for each vaccine, which may affect the immune response.  The mention of adjuvant is also relevant here, which must be taken into context when stating that one vaccination method is better than another (example here for IFN response).

Line 220: Table 1 is not referred to in the text

Line 226: The authors should mention hemispheres in context of the annual vaccine update, possible reference to WHO.

Line 228, The authors should mention that the “one IBV” strain is that to be predicted to be the dominant circulating lineage for that year.

Line 279 and 284  “H1N1pdm09” typo

Table 2: The authors should reformat all strain names to the full name, full year, eg. A/Wisconsin/67/2005 not A/WS/67/05 or any combination of shortcuts. A mix of different nomenclature is used in this document.

Line 316, 331 and 340 “immunosenescence”

Line 325 space before reference [106]

Line 346, “antibody responses were significantly associated with frailty” Is this a negative association? Perhaps a word was left out.

Line 350, is has been clear for some time that original antigenic sin is implicated in influenza responses from infection as well as vaccination. The “recent” reference should be rephrased.

Line 366, reference for OAS, 10.1084/jem.124.3.331 is a good candidate

Line 375 “how do”

Line 491 “infection”

Line 530: this reference and information presented may strengthen this section https://www.ncbi.nlm.nih.gov/pmc/articles/PMC6009068/

Line 537 “The method of antigen..”

Line 542, The authors should add a short comment that this is due to adaptation to eggs or cold growing temperatures. Example paper: https://www.ncbi.nlm.nih.gov/pmc/articles/PMC5394856/  

Section 7: This section requires work on NA assays (ELLA, MUNANA) as well as mention of pseudotype based assays for influenza. As mRNA has fast become the leading and promising SARS-CoV-2 vaccine delivery platform that will no doubt be used for flu – pseudotypes have also become incredibly popular and will be used extensively for influenza in the years to come. It is worth mentioning these assays to complete the review.  It is also worth a mention of SRID/SRH that is still considered gold standard in many settings.

No mention has been made of the samples used in these assays (serum, plasma, species), and the treatments (heat inactivation, RDE treatment), this should be rectified as these are important considerations.

Line 566, remove comma after “vaccination, to it.”

Figure 1: If possible, the resolution should be improved before publication.

Line 585 “preparing” not “pre-paring”

References: while not a Viruses requirement, I would recommend to add the DOI of each paper in the reference section.

Author Response

Point 1: Line 49: “majority of antigenic sites” is too general, the authors should be more specific in stating that the majority of antibody responses in humans are directed to antigens in the globular head.

Response 1: The sentence was clarified to “The globular head contains the receptor binding sites and most of the human antibody responses are directed towards epitopes located here.” Line 50.

Point 2: Line 53-54: “Immunity mediated by the HA globular head…” This statement is slightly misleading in the current wording.  Please be more specific such as “may even provide lifelong protective immunity against antigenically-related viruses that have not undergone antigenic drift.”

Response 2: The sentence was clarified to “Immunity mediated by the HA globular head-antibodies following natural infection may even provide lifelong protective immunity against antigenically-related viruses that have not undergone antigenic drift.” Line 54-57.

Point 3: Line 59: The authors should mention HAI correlates of protection here, as despite its drawback the HAI assay is still gold standard, as referenced on line 106 [33]

Response 3: Added “…. and are considered to be an established correlate of protection in healthy adults”. Line 60.

Point 4: Line 71 “not generally neutralizing”, this could be clarified by saying “Antibodies targeting NA are not generally able to inhibit virus entry, rather they have been shown to limit viral egress”. This clarifies that an antibody to NA is not neutralising, but inhibiting enzymatic activity which has an impact on a different part of the virus life cycle.

Response 4: “not generally neutralizing” was changed to “Antibodies targeting NA are previously thought to be unable to inhibit virus entry, and just prevent viral egress [11], but a recent study showed otherwise [16].” Line 72-74.

Point 5: Line 82: The authors should include paper on tetrameric NA, therefore NA quality being crucial for immune responses. Eg. https://www.ncbi.nlm.nih.gov/pmc/articles/PMC5885027/

Response 5: Addedand tetrameric NA activity” and reference by Krammer et al. Line 86.

Point 6: Line 101, “virus neutralization” as a subtitle is deceiving when talking about HAI research

Response 6: We thank the reviewer for this comment. We intended to highlight the role of virus neutralization as an immune mechanism of protection. We may have inadvertently placed too much emphasis on HAI-research instead. We have added additional sentences to clarify that virus neutralization can also be mediated by non-HAI antibodies. “Aside from the HA globular head, antibodies targeting other viral epitopes can also inhibit influenza virus infections in vitro and show protection in animal models. As de-scribed above, broadly-neutralizing influenza antibodies typically target conserved HA and NA epitopes, although may not be elicited in high titers after infection [38]. Dugan et al., recently showed that monoclonal antibodies targeting NA but not NP had neutralization activity and were protective in virus-challenged mice. However, their neutralization and protective potency were reduced compared to HA-globular head or HA-stem antibodies [16].” Line 117-124:

Point 7: Line 177 “immune correlates of robust antibody response” is not well used in the literature and could be misleading. This should possibly be changed to “factors influencing robust antibody response” or a similar type of statement excluding “immune correlates”

Response 7: “immune correlates of robust antibody response” was changed to “factors influencing robust antibody response”. Line 177.

Point 8: Line 183 to 185: The authors should consider mentioning the delivery route for each vaccine, which may affect the immune response.  The mention of adjuvant is also relevant here, which must be taken into context when stating that one vaccination method is better than another (example here for IFN response).

Response 8: We have clarified this sentence by mentioning sampling site and the influence of delivery route (in bold). The modified sentence reads as “Although a more rapid IFN-response in the peripheral blood was reported for TIV, it is unclear how much of this was influenced the route of vaccine administration [67].” Line 183 to 185.

Point 9: Line 220: Table 1 is not referred to in the text

Response 9: Added “A summary of representative studies and their key findings are shown in Table 1” in the text. Line 226.

Point 10: Line 226: The authors should mention hemispheres in context of the annual vaccine update, possible reference to WHO.

Response 10: We have added the specific hemispheres (in bold) and the WHO reference to this sentence “Although there are many licensed formulations for seasonal influenza vaccines, the two most established are the Inactivated Influenza Vaccine (IIV) and LAIV, which contain either three (trivalent, TIV) or four (quadrivalent, QIV) strains and are updated annually for the Northern or Southern Hemisphere based on the strains most likely to predominate in the upcoming influenza season [83]”. Line 232-236

Point 11: Line 228, The authors should mention that the “one IBV” strain is that to be predicted to be the dominant circulating lineage for that year.

Response 11: It has been modified to include “which was predicted to be the dominant circulating lineage for that year”. It now reads as: “TIV and QIV contain two subtypes of IAV, A(H1N1) and A(H3N2), TIV contains one IBV, which was predicted to be the dominant circulating lineage for that year, while QIV will contain two IBVs from both lineages (Yamagata and Victoria). Line 236-239.

Point 12: Line 279 and 284  “H1N1pdm09” typo

Response 12: This error was corrected.

Point 13: Table 2: The authors should reformat all strain names to the full name, full year, eg. A/Wisconsin/67/2005 not A/WS/67/05 or any combination of shortcuts. A mix of different nomenclature is used in this document.

Response 13: All strain names were changed in the column of ‘Subtype’ but not in the column of ‘Study’ since that listed the original titles of the publications.

Point 14: Line 316, 331 and 340 “immunosenescence”

Response 14: This mistake was corrected.

Point 15: Line 325 space before reference [106]

Response 15: This mistake was corrected.

Point 16: Line 346, “antibody responses were significantly associated with frailty” Is this a negative association? Perhaps a word was left out.

Response 16: The sentence was clarified to “antibody responses were positively associated with frailty”.

Point 17: Line 350, is has been clear for some time that original antigenic sin is implicated in influenza responses from infection as well as vaccination. The “recent” reference should be rephrased.

Response 17: Line 350-We thank the reviewer for raising this point. We have clarified the sentence to more accurately express what we wanted to convey, which was to highlight the different ways pre-existing immunity can influence downstream antibody response. The sentence now reads: “It is now becoming clear that pre-existing influenza immunity can influence the antibody responses after influenza vaccination and infection in different ways.”  Line 354.

Point 18: Line 366, reference for OAS, 10.1084/jem.124.3.331 is a good candidate

Response 18: We thank the reviewer for pointing out this oversight. We have added the reference by Groth et.at.

Point 19: Line 375 “how do”

Response 19: This mistake was corrected.

Point 20: Line 491 “infection”

Response 20: This mistake was corrected.

Point 21: Line 530: this reference and information presented may strengthen this section https://www.ncbi.nlm.nih.gov/pmc/articles/PMC6009068/

Response 21: Thank you for this suggestion. We have added the reference by Trombetta et. al.

Point 22: Line 537 “The method of antigen..”

Response 22: This error was corrected.

Point 23: Line 542, The authors should add a short comment that this is due to adaptation to eggs or cold growing temperatures. Example paper: https://www.ncbi.nlm.nih.gov/pmc/articles/PMC5394856/ 

Response 23: We have added a comment regarding the above in Line 569. The revised sentence reads as: “Egg-grown influenza viruses used in production of most influenza vaccines can also be antigenically distinct from cell-grown viruses, which are more similar to circulating strains [221], since egg grown viruses can acquire egg-adaptive mutations that impacts antigenicity [222, 223].”

Point 24: Section 7: This section requires work on NA assays (ELLA, MUNANA) as well as mention of pseudotype based assays for influenza. As mRNA has fast become the leading and promising SARS-CoV-2 vaccine delivery platform that will no doubt be used for flu – pseudotypes have also become incredibly popular and will be used extensively for influenza in the years to come. It is worth mentioning these assays to complete the review.  It is also worth a mention of SRID/SRH that is still considered gold standard in many settings.

No mention has been made of the samples used in these assays (serum, plasma, species), and the treatments (heat inactivation, RDE treatment), this should be rectified as these are important considerations.

Response 24: We thank the reviewer for the suggestion. We have expanded the discussion in this section to include the above assays as below:

- “Although not as commonly used, single-radial hemolysis (SRH) is are also considered a gold-standard in influenza immunogenicity evaluations by some regulatory bodies. Because it relies on complements to cause hemolysis, SRH can be used to detect com-plement activating influenza antibodies. These serological assays, along with ELISAs are capable of detecting influenza antibodies with different functional property and generally show good correlations to one another in a robust antibody response [218, 219]..” Line 559 to 565.

- “The commonly used method to measure NAI antibodies is the enzyme-linked lectin assay (ELLA), which was originally developed by Lambre et al. [225]. It measures the degree of serum inhibition of NA-ezymatic activity using fetuin as a substrate. Compared with the traditional thiobarbituric acid (TBA) method described by Webster and Laver [226], this method offered the advantages of being scaleable, safe and specific [227]. Another NAI assay relies on the enzymatic cleavage of smaller substrates; either the fluorescent 2′-(4-Methylumbelliferyl)-α-D-N-acetylneuraminic acid (MUNANA) assay [228] or the chemiluminescent NA-STAR assay [229]. Because of the small substrates, NAI-antibodies detected using the MUNANA or NA-STAR assays are thought to only bind near the enzyme’s active site.” Line 578-589.

- “It should also be mentioned that new assays are being developed to detect influenza antibodies. One such approach is the use of pseudoviruses bearing target influenza an-tigens. There are now many systems employed to produce these pseudoviruses [230]. The major advantages of using pseudoviruses are its safety, which is particularly relevant when expressing antigens from highly-pathogenic avian influenza viruses, and its ability to express specific peptides, such as the HA-stem and M2. This enables detection of an-tibodies targeting these epitopes that are normally inaccessible on wild-type viruses. The challenge remains in validating and standardizing these assays to current serological standards as they have shown variable performance in several comparative studies [219, 231].” Line 587-596.

- “Serum is considered as the best specimen of choice for serologic assays. Compared to serum, plasma contained anticoagulants that can sometimes interfere with anti-body-antigen interaction. HI titers were higher against influenza B in plasma, causing the overestimation and underestimation of the seropositive rates [233]. In general however, there is a high degree correlation for HI and neutralization assays against influenza A between serum and plasma, which indicated that plasma can be used as an alternative specimen of choice for these assays where convenient or necessary [233, 234]. Appropriate sample treatment to remove non-specific inhibitors in human and animal sera is also crucial for accurate results. For HI assay, several treatment protocols are available alt-hough none of it worked universally across strains and species [235]. Treatment with receptor destroying enzyme (RDE) from Vibrio cholerae and heat-inactivation is presently recommended as part of the standard WHO HI protocol [236]. For an in-depth review on the serologic assays for influenza, we recommend reference [237].” Line 603-615

Point 25: Line 566, remove comma after “vaccination, to it.”

Response 25: We have removed the comma.

Point 26: Figure 1: If possible, the resolution should be improved before publication.

Response 26: We have provided a new figure with improved resolution.

Point 27: Line 585 “preparing” not “pre-paring”

Response 27: This error was corrected.

Point 28: References: while not a Viruses requirement, I would recommend to add the DOI of each paper in the reference section.

Response 28: We have included DOI in the reference section.

Reviewer 2 Report

It is a comprehensive and nicely written literature study covering a very important topic. One would suggest mentioning briefly on virus protein microarray used for assessment of post-infection immune response (Meade et al., mBio, 2020, 11(1): e03243-19) and also expanding the discussion about preexisting immunity to define the antibody responses from some recent works by the group of Patrick Wilson (e.g., Dugan et al., Sci Transl Med, 2020 12(573): eabd3601). Other than that, the manuscript can be considered for publication in Viruses

Author Response

Point 1: It is a comprehensive and nicely written literature study covering a very important topic. One would suggest mentioning briefly on virus protein microarray used for assessment of post-infection immune response (Meade et al., mBio, 2020, 11(1): e03243-19) and also expanding the discussion about preexisting immunity to define the antibody responses from some recent works by the group of Patrick Wilson (e.g., Dugan et al., Sci Transl Med, 2020 12(573): eabd3601). Other than that, the manuscript can be considered for publication in Viruses.

Response 1: We thank Reviewer 2 for the suggestions. We have expanded our discussion as suggested. Specifically, we have added:

  1. “Indeed, using an influenza virus protein microarray, Meade et al. found that children under 6 years of age had a narrow lgG and lgA antibody response while adults showed a broad recall response [137].” in line 410-412.
  2. “A recent informative study by Dugan et al., characterized the monoclonal antibodies derived from plasmablasts induced after infection and vaccination and found that only an average of 29% of antibodies elicited after infection have in vitro virus neutralizing activity, compared to 80% of antibodies that were elicited after vaccination. The non-neutralizing antibodies induced after infection targets more conserved epitopes such as HA-stalk, NA and NP and other as yet unidentified epitopes. Interestingly, they noted differences in the H1N1 and H3N2 responses; H3N2 were more likely to elicit cross-reactive antibody re-sponses compared to H1N1, which they attributed to the more rapid evolution of H3N2 viruses [16]. ” in line 385-392.
  3. “Collectively, these studies demonstrate that each new exposure does demonstrate a bias in recalling pre-existing memory B-cells, which with repeated exposures, may be enriched against conserved epitopes that has less potent neutralizing activity.” In line 398-401.